# Alpha 2 agonists for sedation to produce better outcomes from critical illness (A2B Trial): protocol for a multicentre phase 3 pragmatic clinical and cost-effectiveness randomised trial in the UK

Timothy Simon Walsh ,[1] Leanne M Aitken ,[2] Cathrine A McKenzie,[3] Julia Boyd,[4] Alix Macdonald,[1] Annabel Giddings,[1] David Hope,[5] John Norrie ,[6] Christopher Weir ,[7] Richard Anthony Parker,[8] Nazir I Lone,[1] Lydia Emerson,[2] Kalliopi Kydonaki,[9] Benedict Creagh-Brown,[10,11] Stephen Morris ,[12] Daniel Francis McAuley ,[13] Paul Dark,[14] Matt P Wise,[15] Anthony C Gordon,[16] Gavin Perkins,[17] Michael Reade ,[18] Bronagh Blackwood ,[19] Alasdair MacLullich,[20] Robert Glen,[5] Valerie J Page[21,22]

**Correspondence to**
Professor Timothy Simon Walsh;
twalsh@staffmail.ed.ac.uk

## ABSTRACT

**Introduction** Almost all patients receiving mechanical ventilation (MV) in intensive care units (ICUs) require analgesia and sedation. The most widely used sedative drug is propofol, but there is uncertainty whether alpha2-agonists are superior. The alpha 2 agonists for sedation to produce better outcomes from critical illness (A2B) trial aims to determine whether clonidine or dexmedetomidine (or both) are clinically and cost-effective in MV ICU patients compared with usual care.

**Methods and analysis** Adult ICU patients within 48 hours of starting MV, expected to require at least 24 hours further MV, are randomised in an open-label three arm trial to receive propofol (usual care) or clonidine or dexmedetomidine as primary sedative, plus analgesia according to local practice. Exclusions include patients with primary brain injury; postcardiac arrest; other neurological conditions; or bradycardia. Unless clinically contraindicated, sedation is titrated using weight-based dosing guidance to achieve a Richmond-Agitation-Sedation score of −2 or greater as early as considered safe by clinicians. The primary outcome is time to successful extubation. Secondary ICU outcomes include delirium and coma incidence/duration, sedation quality, predefined adverse events, mortality and ICU length of stay. Post-ICU outcomes include mortality, anxiety and depression, post-traumatic stress, cognitive function and health-related quality of life at 6-month follow-up. A process evaluation and health economic evaluation are embedded in the trial. The analytic framework uses a hierarchical approach to maximise efficiency and control type I error. Stage 1 tests whether each alpha2-agonist is superior to propofol. If either/both interventions are superior, stages 2 and 3 testing explores which alpha2-agonist is more effective. To detect a mean difference of 2 days in MV duration, we aim to recruit 1437 patients (479 per group) in 40–50 UK ICUs.

**Ethics and dissemination** The Scotland A REC approved the trial (18/SS/0085). We use a surrogate decision-maker or deferred consent model consistent with UK law. Dissemination will be via publications, presentations and updated guidelines.

**Trial registration number** ClinicalTrials.gov NCT03653832.

## STRENGTHS AND LIMITATIONS OF THIS STUDY

⇒ This is the largest randomised trial simultaneously comparing both clonidine and dexmedetomidine to propofol (usual care) in a pragmatic effectiveness design.

⇒ The trial maximises efficiency by using a hierarchical approach to hypothesis testing that primarily establishes whether each alpha2-agonist is superior to propofol, but retains power to explore their relative effectiveness if this is demonstrated.

⇒ The trial includes a process evaluation that will provide information to help understand the results.

⇒ The trial includes a detailed health economic evaluation, which is relevant because intensive care unit care is costly and there are differences in costs between the drugs which are changing over time.

⇒ The trial only has moderate power to detect potentially important differences in mortality, and heterogeneity of effects according to patient age and other factors.

## INTRODUCTION

Around 20 million patients worldwide require intubation and mechanical ventilation (MV) in intensive care units (ICUs) each year.[1] Almost all require sedation and analgesia to relieve pain and anxiety, achieve comfort and facilitate treatment. Guidelines recommend that patients are kept awake or lightly sedated

whenever possible, and as early during ICU care as possible.[2–4] Sedative choice may influence the prevalence and duration of delirium, which is associated with adverse outcomes. However, it remains uncertain whether this relationship is causal, in part because delirium prevention and management strategies have been ineffective in most studies.

Research has shown an association between deep sedation and adverse short-term outcomes including prolonged MV and ICU stay, hospital-acquired infections and greater mortality, although this evidence has been inconsistent.[2 5 6] A concern regarding keeping patients more awake has been whether long-term psychological morbidity, such as post-traumatic stress, anxiety and depression might be increased.[7–9] It is uncertain whether 'light sedation' strategies or the choice of sedative agent can modify this, either directly or by decreasing delirium.[8 10 11]

The most established drugs for patient sedation are the gamma-aminobutyric acid receptor (GABA) agonists, namely propofol or benzodiazepines. These are prescribed once adequate analgesia, usually with opioid drugs, has been established. Benzodiazepines are associated with greater delirium, and propofol is recommended for first-line use in guidelines and is the first-line sedative in the UK. Alpha2 agonists are an alternative class of sedative that provide sedation by dose-dependent decrease in noradrenergic neuron activity in the brain stem via presynaptic and postsynaptic receptor-mediated effects.[12] Unlike GABAergic sedatives, alpha2 agonists have analgesic properties, which can reduce opioid requirements.[13] Two alpha2-agonists are in widespread use in ICUs in the UK.

*Dexmedetomidine* is a highly selective alpha2-agonist with a α2:α1 receptor selectivity ratio of 1620:1.[14] It was developed as a sedative agent and is licensed for intravenous ICU sedation. The drug is >90% protein bound. Unbound drug crosses the blood-brain barrier to exert central effects. Metabolism in the liver creates inactive metabolites which are excreted renally. Renal impairment does not significantly alter clinical effects. The terminal elimination half-life is around 2 hours.

*Clonidine* was the prototype alpha2-agonist, licensed for hypertension, but subsequently used therapeutically for a wide range of neuropsychiatric conditions, drug withdrawal syndromes, and in pain medicine.[15] The drug is available in multiple formulations (including oral, transdermal and intravenous). Many clinical uses are unlicensed, including ICU sedation via any route. Clonidine has significantly lower α2-receptor selectivity than dexmedetomidine; α2:α1 selectivity is 220:1 (×8 less than dexmedetomidine). Clonidine is less protein bound than dexmedetomidine (20%–40%), and around 65% is excreted unchanged in the urine. The elimination half-life is significantly longer and variable (typically 5–13 hours), and (unlike dexmedetomidine) is prolonged by renal failure (18–41 hours). Peak effects after a single dose occur after 10–60 min, but may last 3–7 hours.

A survey of UK ICUs when planning this trial found 58% of ICUs use dexmedetomidine, but in less than 10% of patients. More than 90% used clonidine, in up to 25% of patients, but administration route and protocols varied widely. Widespread practice variation was present. Although widely used in the UK, intravenous clonidine has limited international use and is not included in international guidelines.[16] Dexmedetomidine is licensed for ICU sedation and has been manufactured 'off patent' since 2019. Clonidine not licensed for ICU use, but is administered via both oral/enteral and intravenous routes, especially for the management of agitation and delirium.

## Current evidence

The safety and effectiveness of clonidine for ICU sedation have not been studied in large randomised trials. A systematic review (SR) of studies in critical care included eight studies (643 patients).[17] There was important and relevant heterogeneity in multiple areas, including the population; routes of administration (six intravenous and two oral); and dosage regimens. In seven of eight trials, clonidine was used for adjunctive rather than stand-alone sedation. Meta-analysis suggested no effect on clinical outcomes but an association with hypotension (RR 3.11; 95% CI 1.64 to 5.87).

Dexmedetomidine has been widely studied, and evidence summarised in a range of SR and meta-analyses. These have varied in terms of population definition (including SRs of all critically ill MV adults, or restricted to older patients or those with sepsis) and also the comparator (including 'usual care sedation' or propofol). The primary outcomes include mortality, duration of MV, and delirium. SRs prior to 2020 did not include data from the largest trial of dexmedetomidine (see below). The most recent SRs compared dexmedetomidine versus other sedative agents[18] or propofol[19] in critically ill MV adults in published trials to 2022. Dexmedetomidine was found to reduce delirium (moderate certainty), the duration of MV (low certainty) and ICU length of stay (low certainty).[18] There was no effect on mortality at 30 days (moderate certainty). Dexmedetomidine increased the risk of bradycardia and hypotension. Authors commented on population heterogeneity, with different risk profiles for key clinical outcomes.

The 'early sedation with dexmedetomidine in critically ill patients' (SPICE III) trial randomised 4000 patients to receive dexmedetomidine or usual care within 12 hours of ICU admission.[20] The primary outcome of mortality was no different between the groups. Patients in the dexmedetomidine group had more ventilator free days (VFDs) and more days free of coma or delirium during 28 days follow-up. The median duration of ventilation in the trial was 3–4 days, and overall dexmedetomidine patients gained one VFD and had one less day of coma/delirium during 28 days follow-up. There were six predefined subgroup analyses. There were no differences in mortality according to baseline illness severity, severity

of oxygenation impairment, geographical region, admission type (operative/non-operative), or sepsis at enrolment. There was a difference in mortality for patients above and below the median patient age. Patients aged <63.7 years who received dexmedetomidine experienced more deaths (mean absolute risk difference 4.4% (95% CI 0.8% to 7.9%)), and patients aged ≥63.7 years experienced fewer deaths (mean absolute risk difference −4.4% (95% CI −8.7% to −0.1%)). This finding was explored in a detailed *post hoc* analysis which confirmed the finding using a range of statistical approaches, but without an explanation for the effect.[21] A cluster analysis suggested that a beneficial effect on mortality may be most marked in operative versus non-operative patients. Based on these data, a caution around increased mortality risk in patients aged ≤65 years was issued in June 2022 by the European Medicine Agency.[22]

## Pharmacoeconomic considerations

There is a cost-difference between the three agents used in the A2B trial, but the cost of dexmedetomidine has decreased substantially since coming off-licence. Current estimates (August 2023) for a typical UK cost for sedating a 70 kg adult receiving MV in the UK are: propofol £15 (€17); dexmedetomidine £22 (€25) and clonidine £8 (€9). Changes in cost, combined with potential effects on clinically important outcomes mean a health economic evaluation of alpha2-agonists is relevant.

## Research commission and funding

The A2B trial was funded as a UK National Institute of Health and Care Research (NIHR) Health Technology Assessment (HTA) Agency commissioned trial (16/93 'alpha-2 agonists for sedation in critical care', 2017). The project brief specifically highlighted the widespread off-licence use of clonidine in the absence of safety and effectiveness evidence. The funder and grant reference number is: 16/93/01.

## Trial registration

The trial is registered on ClinicalTrials.gov (NCT03653832); EudraCT number is 2018-001650-98. This paper is based on protocol version 7.0 (date: 25 April 2023)

## METHODS AND ANALYSIS

The primary hypothesis is that sedation with alpha2-agonists will decrease the time to extubation in adult MV ICU patients compared with propofol (usual care).

## Design

Randomised, parallel-group, allocation concealed, controlled, open-label, phase 3, pragmatic, clinical and cost-effectiveness trial with an internal pilot. After intubating and stabilising patients, we randomise patients (1: 1: 1) as early as possible to receive sedation-analgesia based on clonidine *or* dexmedetomidine *or* to continue propofol (usual care) plus opioid analgesia as required.

## Patients and public involvement (PPI)

Former ICU patients and their relatives were consulted during the application to the NIHR HTA panel in addressing the importance of the research questions, and the design of the study, through participation in focus groups. A former ICU patient (RG) is a coapplicant on the grant and coinvestigator on the trial. The PPI group were consulted when agreeing the primary and secondary outcomes, and played a key role in agreeing the long term outcome measures, the frequency of assessment, and the tools used to collect them. RG is providing advice throughout the trial. In addition, the Trial Steering Group includes an independent lay member.

## Primary objective

To determine whether intravenous sedation with the alpha2-agonist agents, dexmedetomidine or clonidine, can decrease the time to successful extubation from MV among adult critically ill patients.

## Secondary objectives
### Clinical and person-centred objectives
*During ICU stay,* we compare rates and duration of delirium or coma, time to optimum sedation, average sedation depth, the ability of patients to communicate with staff and relatives, the quality of sedation and duration of ICU stay. We also compare safety based on predefined adverse events (AEs) relevant to sedation and alpha2-agonist agents.

*Following discharge from the ICU,* we compare patient outcomes for which sedation and ICU experience may be on the causal pathway, namely patients' memories of their ICU stay, psychological well-being and cognitive function. We will follow up patients for 6 months for survival, health-related quality of life (HRQoL) and healthcare resource use.

### Economic evaluation
We will include a detailed cost-effectiveness analysis from an NHS and personal social services (PSS) perspective.

### Process evaluation
The trial, by necessity, is a complex healthcare intervention trial evaluating different classes of sedative agents that involves multiple healthcare professionals, assessing and delivering multiple agents using a series of interrelated activities guided by bedside flow charts, across multiple sites. Recognising this, and consistent with the MRC complex intervention framework,[23] we include a process evaluation (PE) to explore the processes involved in intervention delivery, and identify factors and the mechanisms of their interaction likely impacting on trial outcomes.

## Outcomes and endpoints
### Primary endpoint
Time to successful extubation post randomisation (hours). This is defined as:

a. For patients with an endotracheal tube: the time of the first extubation that is followed by 48 hours of spontaneous breathing without mechanical support.

b. For patients with a tracheostomy: the start time of the patient's first period of 48 hours of spontaneous breathing, where spontaneous breathing is defined as receiving support not exceeding 5 $cmH_2O$ positive end expiratory pressure (PEEP) or continuous positive airway pressure (CPAP) with ≤5 $cmH_2O$ pressure support above PEEP.

c. For patients who are receiving non-invasive mechanical ventilation: the start time of the patient's first period of 48 hours of spontaneous breathing, defined as receiving support not exceeding 5 $cmH_2O$ CPAP via mask/hood.

### Secondary outcomes

The A2B trial has a range of clinical and patient centred outcomes, which were discussed and approved following a PPI exercise. These are shown in table 1.

### Study population

The target population are critically ill patients requiring MV, recruited as early during ICU stay as possible, with an anticipated total requirement for MV of *at least* 2 days. Alpha2-agonists are not appropriate as single agents for intubation and early sedation for most acutely ill patients. Anaesthesia to undertake endotracheal intubation and establish initial ICU sedation-analgesia follows current usual care.

### Inclusion and exclusion criteria

Inclusion and exclusion criteria are listed in box 1.

### Screening and consent

Participants are identified by clinical and research teams. Potential participants lack mental capacity. Appropriate approaches to consent according to UK law are used, approaching Personal and Professional legal representatives. The use of the 'emergency provision' can be used for deferred consent when a legal representative is not available within 2 hours of meeting eligibility criteria. In all cases, when patients regain capacity, they are approached for consent to continue in the trial (see online supplemental file 1).

### Randomisation

Randomisation is undertaken immediately after consent is obtained or when deferred consent is triggered by the research team, using a remote web-based randomisation system. Randomisation is in a 1:1:1 ratio to the three interventions using permuted blocks (randomly arranged sizes of 3, 6, 9, 12) stratified by centre. The allocation sequence was generated by a clinical trials unit programmer not involved in clinical management and is stored on a remote secure server concealed from all personnel involved in the trial.

### Intervention groups

Patients commence intravenous infusion of open-label study drug according to a weight-based dose regimen (see online supplemental file 1) as early as possible post randomisation, and within a maximum of 2 hours.

Bedside clinical staff transition patients to achieve sedation with the allocated alpha2-agonist agent as quickly as clinically feasible and safe, using bedside guidance algorithms (see online supplemental file 1). Additional opioid is used for analgesia using clinical judgement. Once alpha2-agonist is established, additional propofol is only recommended when the maximum alpha2-agonist dose is reached or because cardiovascular or other side-effects limit dose escalation.

### Dexmedetomidine group

For dexmedetomidine, starting dose is 0.7 µg/kg/hour titrated to a maximum dose 1.4 µg/kg/hour as per manufacturer guidance. Lower starting doses are used at clinical discretion for patients with cardiovascular instability, for example, for patients on high doses of norepinephrine. No loading dose is administered.

### Clonidine group

For clonidine, the regimen is designed to be equipotent with dexmedetomidine based on known pharmacokinetics and pharmacodynamics. The chosen regimen is similar to that currently used in many UK ICUs as part of routine 'off label' practice. The starting dose is 1.0 µg/kg/hour titrated to a maximum dose of 2 µg/kg/hour. Lower starting doses can be used at clinical discretion for patients with cardiovascular instability as for dexmedetomidine. No loading dose is administered.

### Usual care group

Patients continue to receive intravenous propofol according to current usual care. The sedation targets, weaning and sedation discontinuation procedures follow the same clinical targets as for the intervention groups.

The dosing guidance algorithms are included in the online supplemental file 1.

### Duration of intervention

The intervention period continues until: (1) The patient is successfully extubated according to the definition of the primary outcome; or (2) the patient dies during MV in the ICU; or (3) the patient is transferred to another non-participating ICU prior to achieving the primary outcome, or (4) 28 days of MV in ICU have been required following randomisation without achieving the primary outcome.

Timing of discontinuation of sedative agents is at the discretion of the clinical team. If the patient is reintubated before achieving the primary outcome, they continue with group allocated treatment until the primary outcome is successfully achieved.

### Management during the intervention period

The default sedation target is the most awake and comfortable state considered safe by clinical staff. For each 12 hours nursing shift, clinical staff document whether there is a clinical indication for deep sedation, such as brain

**Table 1** Secondary outcomes, measurement tool or method, and timing

| Outcome | Measurement tool or method | Timing |
| --- | --- | --- |
| **Mortality** | Medical records check | ICU, hospital, 30, 90 and 180 days post randomisation |
| **Length of ICU stay**<br>Number of days the participant is in ICU | Medical record | ICU discharge |
| **Sedation and analgesia quality**<br>Lowest and highest RASS score per day over time during intervention<br>Quality of sedation using SQAT states (daily basis); days with optimum sedation, agitation, or unnecessary deep sedation (RASS -4/–5).<br>Quality of analgesia using presence of pain behaviour (daily basis) based on limb response to movement and ventilation compliance | Richmond Agitation and Sedation Scale (RASS)<br>Sedation Quality (based on Sedation Quality Assessment Tool (SQAT)).[27]<br>Two components of the SQAT pain assessment will be used in this trial to measure sedation quality (limb relaxation and compliance with ventilation)<br>Defines four states for sedation quality:<br>1. Overall optimum sedation (no agitation; no unnecessary deep sedation; no pain behaviour)<br>2. Agitation<br>3. Unnecessary deep sedation (RASS -4/–5 without clinical indication)<br>4. Pain (presence of pain behaviour based on limb response to movement and ventilation compliance) | Four hourly during ICU stay until primary outcome is reached<br>Derived from daily sedation and analgesia quality data during intervention period in ICU until primary outcome is reached |
| **Time to first optimum sedation** hours<br>Hours from randomisation to first 'light' sedation (RASS score of –2 or greater)<br>Days from randomisation to first day with optimum sedation (based on SQAT definition) | RASS scores 4 hourly during ICU stay<br>SQAT status (daily during ICU stay) | Based on daily sedation and pain assessments during the intervention period |
| **Delirium prior to successful extubation**<br>Occurrence prior to successful extubation (binary outcome)<br>Days with delirium (CAM-ICU positive) or coma (RASS score -4/–5) prior to successful extubation (continuous outcome) | Confusion Assessment Method for the ICU (CAM-ICU)[28] | Twice daily during ICU stay until primary outcome is reached |
| **Drug-related adverse events**<br>Number of patients experiencing a predefined adverse event and each defined adverse event<br>Number of days prior to successful extubation that any predefined adverse event occurred, and each defined adverse event occurred. | Severe bradycardia; cardiac arrhythmias; cardiac arrest (defined in protocol); ileus | Daily during the intervention period |
| **Health-related quality of Life**<br>HRQoL at 30, 90 and 180 days post randomisation | EuroQol tool (EQ-5D-5L) | Recalled HRQoL prior to hospital admission; prospective measurement 30, 90 and 180 days post randomisation |
| **Patients' ability to communicate pain and ability to cooperate with care**<br>Number of days on which pain could be communicated during intervention (binary score)<br>Number of days on which patient was able to cooperate with care (binary score) | Binary assessment for each 12 hours nursing shift requested from bedside nurse (based on overall assessment of period of care). Answer to the following questions:<br>1. Was your patient able to communicate pain?<br>2. Was your patient able to cooperate with care? | Twice daily until primary outcome is reached |
| **Patient experience of ICU care**<br>ICE-Q score at 90 days post-randomisation overall for each domain | Intensive Care Experience Questionnaire (ICE-Q)[29]<br>Provides numeric score in four domains:<br>1. Awareness of surroundings<br>2. Frightening experiences<br>3. Recall of experiences<br>4. Satisfaction with care | 90 days post randomisation |

Continued

**Table 1** Continued

| Outcome | Measurement tool or method | Timing |
|---|---|---|
| **Relative/partner/friend (*PerLR*) assessment of comfort and communication** Daily response to each of the three questions (binary outcome) | Relative/partner/friends response to the following questions (based on their opinion at time of assessment): 1. Does the patient appear awake to the visitor? 2. Does the patient seem comfortable to the visitor? 3. Does the visitor feel they can communicate with the patient? | Daily at a visit until primary outcome is reached |
| **Anxiety and depression** HADS score at 180 days post randomisation | Hospital Anxiety and Depression Scale (HADS) questionnaire | 180 days post randomisation |
| **Post-traumatic stress** Impact of Events Scale-revised (IES-R) score at 180 days post-randomisation | Impact of Events Scale-revised (IES-R) | 180 days post randomisation |
| **Cognitive function** TMoCA score at 180 days post randomisation | Montreal Cognitive Assessment Tool (Telephone version) (TMoCA) | 180 days post randomisation |

ICU, intensive care unit.

injury, seizures or a requirement for advanced MV modes. If deep sedation is required, the allocated sedative agent is titrated to achieve this if feasible. In the absence of clinical requirement for deep sedation, the *least awake* target sedation state will be 'brief eye contact made in response to voice' (Richmond Agitation and Sedation Scale (RASS) score of −2). The additional use of daily sedation breaks is at the discretion of the caring clinical teams.

Staff in participating ICUs receive training in the trial protocol prior to recruiting patients. RASS score is recorded every 4 hours. The bedside algorithms recommend changes to sedation drug (according to group allocation) based on responses to RASS scores (see online supplemental file 1). Patients receive opioid infusions for analgesia as clinically indicated. Patients who require additional sedation or treatment, for example, for agitation, receive this according to local practice.

Patients receiving norepinephrine or other vasopressors at enrolment can be commenced on lower doses of alpha2-agonist. This is suggested when the dose of norepinephrine is more than 0.15 µg/kg/min. Patients who develop hypotension and/or bradycardia in any treatment group are managed according to local practices using fluids and/or vasopressors. Sedative drugs can be reduced or stopped based on clinical discretion. In the alpha2-agonist groups, if the patient's heart rate decreases to less than 50/minute, the alpha2-agonist is stopped until the heart rate increases to greater than 50/min. Restarting the allocated sedative regimen is encouraged once cardiovascular instability has improved.

### Weaning from MV
All patients have regular assessments and attempts to wean and discontinue MV throughout treatment. The approach used in individual ICUs and patients should adhere to 'best practice' principles for weaning from MV. The protocol does not control decisions about weaning sedation and MV tightly, given the pragmatic effectiveness design. Decisions and their timing are at the discretion of the responsible clinical team.

### Data collection
Data collection throughout the study is shown in table 2. Study data are recorded into a case report form (CRF), and transcribed into the web-based electronic CRF within the Edinburgh Clinical Trials Unit (ECTU). Automated query identification and checking is managed and resolved by the trial management team. A trial monitoring strategy by the sponsor tracks data quality at sites and triggers any corrective actions.

### Withdrawals
Participants or their relatives can withdraw at any time. The three options for ongoing data collection will be: withdraw from intervention only, but follow-up and all data collection continues; intervention and follow-up only, with collection of routine data allowed; or withdrawal from all aspects of the trial and follow-up. Wherever possible, primary outcome data are recorded for any withdrawn patient.

### Design and analysis plan
#### Analytic framework
The hierarchical analytic framework was devised to address key clinical effectiveness questions in a staged manner, to enable an efficient trial design that controls overall 'family-wise' type 1 error rate. The trial will determine whether alpha2-agonists are superior to current practice but also, if superiority is found, *which* agent is more clinically effective. We propose three analytic stages, where progression to hypothesis testing in sequential stages is dependent on preceding results (see figure 1). A detailed justification and explanation of these stages is

## Box 1 Inclusion and exclusion criteria for the A2B trial.

### Inclusion criteria

1. Patient requiring mechanical ventilation (MV) in an intensive care unit (ICU)
2. Aged 18 or over
3. Within 48 hours of first episode of MV in ICU
4. Requiring sedation with propofol
5. Expected to require *a total* of 48 hours of MV or more in ICU
6. Expected to require a further 24 hours of MV or more *at the time of randomisation* in the opinion of the responsible clinician

### Exclusions

1. Acute brain injury (traumatic brain injury; intracranial haemorrhage; ischaemic brain injury from stroke or hypoperfusion)*
2. Postcardiac arrest (where there is clinical concern about hypoxic brain injury)*
3. Status epilepticus*
4. Continuous therapeutic neuromuscular paralysis at the time of screening or randomisation*
5. Guillain-Barre Syndrome*
6. Myasthenia gravis*
7. Home ventilation*†
8. Fulminant hepatic failure‡
9. Patient not expected by responsible clinician to survive 24 hours
10. Decision to provide only palliative or end-of-life care
11. Pregnancy
12. Known allergy to one of the study drugs
13. Patient known to have experienced a period with heart rate<50 beats/min for 60 min or longer since commencing MV in the ICU
14. Untreated second or third degree heart block§
15. Transferred from another ICU in which MV occurred for >6 hours
16. Prisoners
17. Enrolled on another Clinical Trial of an Investigational Medicinal Product
18. Previously enrolled on the A2B Trial

Note: Criteria 5 and 6 are intended to ensure that all participants require at least 48 hours of MV in the ICU and that all patients receive at least 24 hours of the allocated intervention after randomisation.
Note:
*For these conditions, the neuromuscular condition will dominate the primary outcome unrelated to sedation practice.
†Home ventilation does not include patients receiving night-time continuous positive airway pressure and/or Bilevel Positive Airway Pressure ventilation (BIPAP) therapy for the treatment of obstructive sleep apnoea syndrome.
‡Uncertain pharmacokinetics of $\alpha-2$ agonist; potential for cerebral oedema mandating deep sedation.
§Patients with treated heart block, for example, with a pacemaker, are eligible for inclusion.

included in the statistical analysis plan (SAP; see online supplemental file 2).

Further details regarding the original rationale for the study design and formation of the sample size calculations have been presented elsewhere.[24]

### Power and sample size during trial design

Based on clinical consensus, likely economic benefit, and the findings of SRs, a minimum clinically important difference of a mean difference in MV of 2 days was chosen for all superiority tests. For non-inferiority of clonidine versus dexmedetomidine, a non-inferiority margin of 1 day was chosen.

Sample size and power were modelled based on the analytic framework outlined in figure 1, which includes a hierarchical approach to hypothesis testing to control the 'familywise' type I error to 5%. We used a large prospective data set from a sedation trial in 8 UK ICUs for modelling (n=708).[25] Based on this data set, we estimate that 53% of patients in the 'usual care' group will be extubated and around 14% will have died prior to extubation at 7 days.

*Stage 1:* if either dexmedetomidine or clonidine are superior to usual care by an overall mean difference of 2 days in time to extubation, this translates to an estimated extubation rate of 63% in the dexmedetomidine or clonidine arm at 7 days. The death rate of 14% was assumed to remain the same as for the usual care arm. Under these conditions, using nQuery V.8 software (log-rank test accounting for competing risks), a sample size of 550 per arm (1650 patients in total, 1328 extubation events across the three arms) has 99% power to detect HRs of 1.37 indicating superiority of clonidine or dexmedetomidine over usual care, assuming a one-sided 2.5% significance level.

*Stage 2:* these analyses are only undertaken if one or other or both of the stage 1 tests are significant. For the non-inferiority test of clonidine relative to dexmedetomidine (test H3), the non-inferiority margin is a 1-day absolute mean difference in time to extubation. Based on the modelled dataset, a 1-day absolute mean difference translates into an estimated probability of 63% in the dexmedetomidine arm and 57% in the clonidine arm achieving the primary outcome at 7 days. This equates to an estimated non-inferiority margin on the HR scale of 0.83, assuming death rates in both arms are 14% at 7 days. Using this information in nQuery V.8 software (log-rank test accounting for competing risks), 550 patients per arm (1100 in total, 888 extubation events) provides 81% power to conclude non-inferiority of clonidine, using a one-sided 2.5% significance level. The power calculation for the superiority comparison of dexmedetomidine vs clonidine (test H4) is the same as that for stage 1. Simulation work was used to calculate the overall power of test H1 (clonidine superiority test vs propofol) *and* test H3 (clonidine non-inferiority test vs dexmedetomidine) being statistically significant using Fine and Gray proportional subdistribution hazards regression analysis based on 2000 trials simulated from the real ICU dataset (mean 7 days on ventilation).[25] Assuming that dexmedetomidine and clonidine are both superior to usual care by an overall true mean difference of 2 days, and there is no difference between dexmedetomidine and clonidine, then a total sample size of 1650 (550 per group) provides 81% power of concluding non-inferiority of clonidine over dexmedetomidine (test H3) *and* concluding clonidine is superior to usual care (test H1) based on simulation, using a one-sided 2.5% significance level.

*Stage 3:* the power calculation for the superiority comparison of clonidine versus dexmedetomidine (test H5), which

**Table 2** Assessments and measurements undertaken during the trial

| | Pre randomisation | Baseline data | Daily ICU data collection* | ICU discharge* | Hospital discharge* | 30 days† | 90 days† | 180 days† |
|---|---|---|---|---|---|---|---|---|
| Screening for eligibility and consent, demographics, CHI/hospital number, RASS, CAM-ICU, final eligibility check | X | | | | | | | |
| Baseline data collection—baseline data, Functional Comorbidity Index (FCI), Acute Physiology and Chronic Health Evaluation (APACHE) II, Sequential Organ Failure Assessment (SOFA), RASS, CAM-ICU, PRE-DELIRIC (collected at 24 hours), EQ-5D-5L (assessed by proxy). | | X | | | | | | |
| Sepsis substudy only—two blood samples for inflammatory markers ▲ Baseline sample (within 12 hours post randomisation) ▲ 60 hour sample (within 48–72 hours post randomisation) | | X | | | | | | |
| Daily data collection during ICU stay until primary outcome confirmed or day 28—clinical team (4 hrly—RASS score and pain assessment; 12 hrly—CAM-ICU, SQAT, cooperation and communication assessment) | | | X | | | | | |
| Daily data collection during ICU stay until primary outcome confirmed or day 28—research team (MV data collection, IMP and drug usage, SOFA score, adverse event data collection) | | | X | | | | | |
| Assessment of comfort and communication by informant until primary outcome confirmed or day 28 | | | X | | | | | |
| Adverse event data collection until ICU discharge | | | X | | | | | |
| ICU and hospital discharge data | | | | X | X | | | |
| Mortality | | | X | X | X | X | X | X |
| Intensive Care Experience Questionnaire (ICE-Q) | | | | | | | X | |
| Hospital Anxiety and Depression Scale (HADS) questionnaire | | | | | | | | X |
| Impact of Events Scale—revised (IES-R) | | | | | | | | X |
| Montreal Cognitive Assessment Tool (Telephone version—TMoCA) | | | | | | | | X |
| Euroqol tool (EQ-5D-5L) | | | | | | X | X | X |
| Recalled Euroqol tool (EQ-5D-5L) | | | | | | X | | |
| Health economic questionnaire (including hospital resource use and return to employment) | | | | | | | X | X |

*These data are collected from the routine health record, except for the EG-5D-5L which is collected from the patient's proxy

†These data are collected by research staff. Site teams confirm patient status, and then the research team contacts the patient using a mixed strategy including postal and telephone contact to maximise completion.

CAM-ICU, Confusion Assessment Method for the ICU; ICU, intensive care unit; MV, mechanical ventilation; PRE-DELIRIC, PREdiction of DELIRium in ICu patients; RASS, Richmond Agitation and Sedation Scale; SQAT, Sedation Quality Assessment Tool.

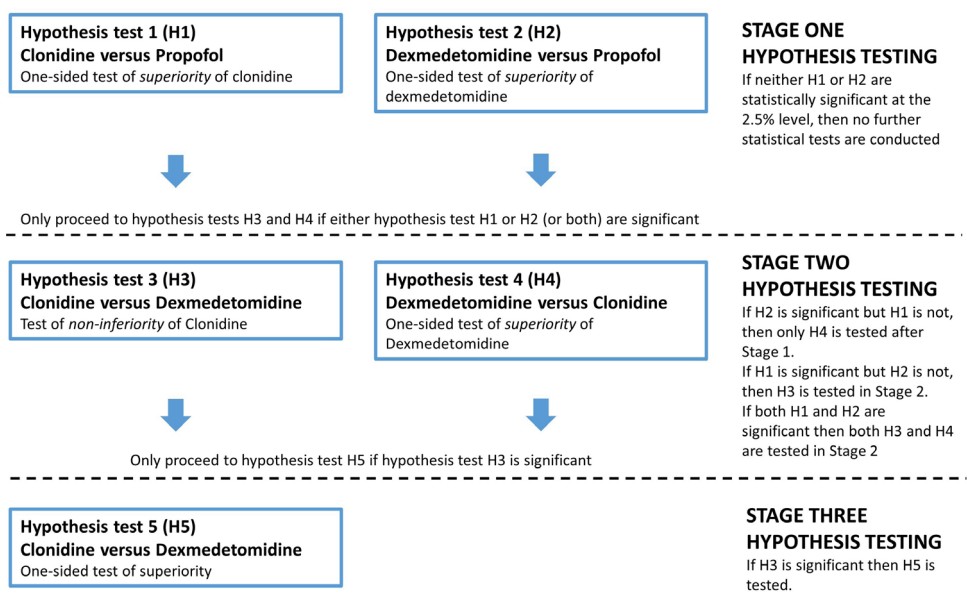

**Figure 1** Hierarchical design and analytics framework used in the A2B trial. Note: all hypothesis tests performed using a one-sided 2.5% significance level in the original design.

will only be done if stage 1 demonstrates superiority (tests H1 or H2) *and* clonidine is non-inferior to dexmedetomidine (test H3), is the same as that given in stage 1.

### Original sample size

We inflated sample size by 5% for loss to follow-up for the primary outcome. The original trial sample size was therefore 1737 (579 patients per group).

### Mortality

For the key outcome of mortality in ICU prior to extubation, a sample size of 550 per group provides 83% power to detect a difference in mortality of 7% (equivalent to a HR of approximately 1.5) using Cox regression assuming mortality in the usual care group is 23% and 16% in the clonidine/dexmedetomidine group, using a two-sided 5% significance level.

### Modifications to sample size due to impact of COVID-19 pandemic

The COVID-19 pandemic had a major impact on the trial progress and recruitment. In consultation with the funder, a modification to the original sample size was agreed in February 2023. The focus was on maintaining high power for the stage 1 hypothesis testing, and included modelling the impact of a reduced sample size on the stage 2 test of non-inferiority of clonidine versus dexmedetomidine, plus the power for detecting an effect on mortality. Based on these investigations, the sample size was reduced to 1437. This maintained 99% power for the stage 1 comparisons of clonidine and dexmedetomidine versus propofol (H1 and H2), and also for the superiority comparison of dexmedetomidine versus clonidine if progression to stage 2 testing occurs (H4). The main effect on power was for the non-inferiority comparison of clonidine versus dexmedetomidine (H3). For this comparison, in order to maintain 80% power when using the non-inferiority margin of 1 day, the significance level for test H3 was increased from 2.5% to 4%. This change to the

hypothesis testing hierarchy meant that the upper limit on the familywise type I error rate increased from 5% to 6.5%. For the key secondary outcome of mortality, for the same 7% mortality difference, power decreased from 83% to 76%.

### Predefined subgroup analyses

We plan four exploratory subgroup analyses, for patients with: (1) sepsis at enrolment; (2) higher delirium risk as defined by the PRE-DELIRIC delirium risk prediction score, using the version assessed at 24 hours post admission[26]; (3) greater organ dysfunction, as measured by SOFA score, at randomisation (as this could differentially alter the safety profile of the three groups); and (4) age≥64 years versus age<64 years (based on the relationship between age and mortality seen in the SPICE III trial).[20 21]

### Statistical analysis plan

An estimand was developed to deal how key intercurrent events will be dealt with in the analysis (see online supplemental file 1). A detailed SAP has been finalised. The current version is included as an electronic supplement (see online supplemental file 2). The most up-to-date version can be found in the statistics section of the Trial Master File held in the ECTU.

### Process evaluation

A PE is included recognising that ICU sedation is a complex healthcare intervention that involves multiple healthcare professionals, assessing and delivering multiple agents using a series of interrelated activities, across multiple sites. The PE aims to: establish the extent to which the intervention is delivered as intended (fidelity, dose and reach), over time and across different ICUs; ascertain how clinical staff understand and respond to the intervention, over time and across different ICUs; and, explore the importance of context (inter-ICU differences, changes over time) and determine factors

(including organisational structure and processes) that affect intervention implementation and delivery. The detailed PE methods and analytic framework will be published separately.

## Health economic evaluation

We will undertake a detailed analysis of the cost-effectiveness of dexmedetomidine, clonidine and usual care. We will estimate costs and cost-effectiveness for both the 'within-trial' period and over the expected lifetime of the patient. Costs will be assessed from the perspective of the NHS and PSS. Quality Adjusted Life Years (QALYs) will be calculated based on the HRQoL and mortality data collected during the trial. Details of the health economic evaluation is included in the Online supplemental file 1.

## Monitoring, pharmacovigilance and safety monitoring

Participants are monitored for AEs and serious adverse events (SAEs) until ICU discharge. Recording and reporting of AEs and SAEs will follow the Standard Operating Procedures of the trial sponsor (the Academic and Clinical Central Office for research and Development, Edinburgh (ACCORD)). A trial monitoring plan designed by the study sponsor is in place, which includes study audits at study sites and within the trial management team and is carried out by independent sponsor QA personnel. All protocol amendments and their dissemination are managed according to sponsor SOPs compliant with UK Health Research Authority (HRA) guidance.

## Ethics and dissemination

The trial is classified as a Clinical Trial of an Investigational Medicinal Product (CTIMP). The trial was reviewed and approved by the Scotland A REC (18/SS/0085), which for a CTIMP provides approval across the UK, and the Medicines and Healthcare products Regulatory Agency. Each participating site undertakes local review and issues R&D approval according to UK HRA processes. As the trial involves incapacitated adults, all consent processes comply with the EU clinical trials regulations as written into UK law. Trial results will be disseminated through publications, conference presentations and media engagement. Trial data will be uploaded to the EudraCT database (https://eudract.ema.europa.eu/).

## Trial management and oversight

The trial is coordinated by a Project Management Group, including trial managers and coordinators, clinical investigators and the statistics teams (see author contributions).

A Trial Steering Committee (TSC) is overseeing the conduct and progress of the trial, comprising an independent Chair, a PPI representative and more than 70% independent clinical and methodology experts. All members sign a TSC charter.

An independent Data Monitoring Committee (DMC) is overseeing the safety of participants in the trial with an agreed DMC Charter to determine Terms of Reference. Given the caution around use in younger patients, the DMC is specifically monitoring safety and outcomes in younger versus older patient group throughout the trial.

The trial sponsor is the ACCORD joint research office of the University of Edinburgh and Lothian Health Board (https://www.accord.scot/). Indemnity for participants is provided through joint sponsorship by the University of Edinburgh and NHS Lothian.

All data are managed according to the General Data Protection Regulations.

The funder and sponsor were not involved in design, but reviewed and approved the protocol and amendments. Neither have involvement in analysis, interpretation, or report writing. The sponsor is monitoring the trial.

## Current status

The trial recruited its first patient in December 2018. An internal feasibility pilot was successfully completed, and the funder approved progression to complete the full trial. Recruitment was severely affected by the COVID-19 pandemic, with many sites closed for much of 2020–21. The trial reopened in late 2020, but recruitment was affected by ICU pressures and research capacity during 2021–2022. The funder requested a review of trial status and proposals to complete the trial in August 2022. The modelling work for a revised sample size, and considerations of plans to complete the trial recruitment, was concluded in October 2022. The final plan was approved by the funder and sponsor in February 2023, with a proposed recruitment end date of November 2023. Current protocol is version 7 (25 April 2023).

**Author affiliations**
[1]The University of Edinburgh Usher Institute of Population Health Sciences and Informatics, Edinburgh, UK
[2]City University of London, London, UK
[3]University of Southampton, Southampton, UK
[4]Edinburgh Clinical Trials Unit, The University of Edinburgh Usher Institute of Population Health Sciences and Informatics, Edinburgh, UK
[5]NHS Lothian, Edinburgh, UK
[6]Usher Institute, Edinburgh Clinical Trials Unit, University of Edinburgh No. 9, Bioquarter, Edinburgh, UK
[7]Edinburgh Clinical Trials Unit, Usher Institute, University of Edinburgh, Edinburgh, UK
[8]Edinburgh Clinical Trials Unit, University of Edinburgh, Edinburgh, UK
[9]Edinburgh Napier University, Edinburgh, UK
[10]Royal Surrey County Hospital NHS Foundation Trust, Guildford, UK
[11]Intensive Care Unit, Royal Surrey County Hospital, Guildford, UK
[12]Primary Care Unit, University of Cambridge, Cambridge, UK
[13]Centre for Experimental Medicine, Queen's University Belfast, Belfast, UK
[14]Intensive Care Unit, University of Manchester, Greater Manchester, UK
[15]Department of Adult Critical Care, University Hospital of Wales, Cardiff, UK
[16]Section of Anaesthetics, Pain Medicine and Intensive Care, Imperial College London, London, UK
[17]Clinical Trials Unit, University of Warwick, Birmingham, UK
[18]University of Queensland, Brisbane, Queensland, Australia
[19]Wellcome-Wolfson Institute for Experimental Medicine, Queen's University Belfast, Belfast, UK
[20]Geriatric Medicine Unit, University of Edinburgh, Edinburgh, UK
[21]Intensive Care, West Hertfordshire Hospitals NHS Trust, Watford, UK
[22]Faculty of Medicine, Imperial College London, London, UK

**Contributors** TSW, LMA, JN, CW, RAP, NIL, KK, BC-B, DFM, PD, MPW, ACG, GP, MR, BB, AM, RG and VJP designed the trial and led the funding application. All contributed to writing the detailed protocol. In addition, JB, DH, AG, AM and LE contributed to protocol development, implementation, monitoring and amendments. The Process Evaluation was designed by LMA, LE, KK, BB and TSW. The statistical design was led by RAP, JN and CW. The Health economic evaluation was designed by SM. TSW is Chief Investigator.

**Funding** This work is supported by the NIHR Health Technology Assessment Programme (HTA 16/93/01). The NIHR Clinical Research Network (CRN) supports the trial.

**Disclaimer** The views expressed are those of the authors and not necessarily those of the NIHR or the Department of Health and Social Care.

**Competing interests** None declared.

**Patient and public involvement** Patients and/or the public were involved in the design, or conduct, or reporting, or dissemination plans of this research. Refer to the Methods section for further details.

**Patient consent for publication** Not applicable.

**Provenance and peer review** Not commissioned; externally peer reviewed.

**ORCID iDs**
Timothy Simon Walsh http://orcid.org/0000-0002-3590-8540
Leanne M Aitken http://orcid.org/0000-0001-5722-9090
John Norrie http://orcid.org/0000-0001-9823-9252
Christopher Weir http://orcid.org/0000-0002-6494-4903
Stephen Morris http://orcid.org/0000-0002-5828-3563
Daniel Francis McAuley http://orcid.org/0000-0002-3283-1947
Michael Reade http://orcid.org/0000-0003-1570-0707
Bronagh Blackwood http://orcid.org/0000-0002-4583-5381

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
