## [Reviewer comments · BMJ Open]

ARTICLE DETAILS

TITLE (PROVISIONAL)	Alpha 2 agonists for sedation to produce better outcomes from critical illness (A2B Trial): Protocol for a multicentre phase 3 pragmatic clinical and cost-effectiveness randomised trial in the United Kingdom
AUTHORS	Walsh, Timothy; Aitken, Leanne M; McKenzie, Cathrine; Boyd, Julia; Macdonald, Alix; Giddings, Annabel; Hope, David; Norrie, John; Weir, Christopher; Parker, Richard; Lone, Nazir; Emerson, Lydia; Kydonaki, Kalliopi; Creagh-Brown, Benedict; Morris, Stephen; McAuley, Daniel; Dark, Paul; Wise, Matt; Gordon, Anthony; Perkins, Gavin; Reade, Michael; Blackwood, Bronagh; MacLulich, Alasdair; Glen, Robert; Page, Valerie

VERSION 1 – REVIEW

REVIEWER	Toft, Palle Odense University Hospital, Anaesthesia and Intensive Care
REVIEW RETURNED	25-Sep-2023

GENERAL COMMENTS	The present study is a randomized controlled trial where the authors compare sedation with propofol to sedation with clonidine or dexmedetomidine. The study is absolutely clinically relevant. The desired level of sedation is a RAS score of minus 2 or greater. The authors have to be praised to chose this level of light sedation. With this level of light sedation a daily wake up trial seems unnecessary. The primary outcome is time to successful extubation. A lot of different hospitals are included in this multicenter study. Different hospitals have different ways to treat patients. My primary suggestion for improvement of the study is to introduce a more uniform or homogeneous treatment in different hospitals. If the patient is deeply sedated this would influence the time to successful extubation and the presence of coma and delirium free days. If the patient is deeply sedated the reason should be registered. As it is now it is up to the single doctor to decide if deep sedation is necessary. There should be some common rules to decide if deep sedation is necessary. The time to successful extubation is also influenced by the time sedation is stopped. In the protocol it should be described when to stop sedation. This would introduce a more uniform treatment between different sites. Extubation is up to the discretion of the attending doctor. A more homogeneous treatment would be introduced if there were certain clinical endpoint which should be fulfilled before extubation. What will happen if it is not possible to obtain sufficient sedation with clonidine or dexmedetomidine and propofol is used to supplement the sedation. Will the patient be excluded?. If clonidine or dexmedetomidine sedation is not sufficient it might
--

	be tempting to increase the administration of opioids. In this way the administration of opioids will be increased in these treatment groups. It should be stated that this must be avoided and the administration of opioids ought to be registered.
--	--

VERSION 1 – AUTHOR RESPONSE

Response to reviewer

Reviewer: 1 Dr. Palle Toft, Odense University Hospital

Thank you to Dr Toft for the useful comments on the clarity of the protocol. As this trial was designed in 2018, and the protocol was approved by the funding body (the National Institute of Health Research (NIHR), UK) in conjunction with comprehensive peer review, we provide clarification to the points made. We limited some detail in the manuscript due to limited word count in this publication. However, we have added some additional text where relevant (see below in response to each comment). The full version of the protocol approved by the ethics committee is longer, has more detail, and will be included with all publications arising from the trial once completed

The present study is a randomized controlled trial where the authors compare sedation with propofol to sedation with clonidine or dexmedetomidine. The study is absolutely clinically relevant.

Thank you. As we noted this was a commissioned trial through the NIHR HTA programme as it was deemed important through the UK national research prioritisation process. This was noted on page 8 under the ‘Research Commission and Funding’ section.

The desired level of sedation is a RAS score of minus 2 or greater. The authors have to be praised to choose this level of light sedation. With this level of light sedation a daily wake up trial seems unnecessary.

We agree that a RASS score of -2 or greater is considered optimum sedation in guidelines and based on previous research. This is embedded in the protocol, as described under the ‘Management during the intervention period’ section on page 14. We agree that daily wake up is often not required with this approach. In this pragmatic effectiveness trial, the additional use of daily wake up was at the discretion of the caring clinical team in each ICU. We have added a sentence clarifying this to the management section on page 14: ‘The additional use of daily sedation breaks is at the discretion of the caring clinical teams’.

The primary outcome is time to successful extubation. A lot of different hospitals are included in this multicenter study. Different hospitals have different ways to treat patients. My primary suggestion for improvement of the study is to introduce a more uniform or homogeneous treatment in different hospitals.

This trial is a large pragmatic effectiveness trial conducted in >40 different ICUs across the UK. We agree that there is natural variation in sedation practice across hospitals, and between individual clinicians. Our effectiveness ‘real world’ design acknowledges this, but randomisation should decrease the risk of bias as variation will be balanced across the groups. Our aim is to conduct a ‘real world’ trial so we do not think this will compromise the external validity of our results. In designing the trial we did consider how best to minimise the risk of bias (ie systematic differences between the groups other than due to the interventions). The following approaches are included, while specifically not intending to tightly control practice (which would be closer to an efficacy design and less ‘real world’ evaluation):

1. We provide clear group specific algorithms to titrate sedation to target sedation level. These are

described on page 14, and each algorithm is included in full in the supplementary material which we believe is a clear description of the approach taken. These are the charts placed at the bedside of all patients enrolled in the trial. We have added the sentence: 'Staff in participating ICUs receive training in the trial protocol prior to recruiting patients' for further clarity.

2. We are describing the sedation practice received as part of the analysis, and this is included in the secondary outcomes. This is detailed in table 1. Specifically, we have defined a priori data analysis algorithms based on the daily data collected during each 12 hours nursing shift that will allow us to measure sedation quality and practice. This includes using a measure of sedation quality, the SQAT, which we previously validated (reference 24). Our approaches are detailed in the statistical analysis plan, which is included in the supplementary material.

3. We include a process evaluation in the trial. This will include exploring sedation across the three intervention groups based on interviews with both research staff and bedside nursing staff. This is described briefly (due to word limitations) on page 19. However, we plan to publish a detailed description of the process evaluation design, alongside data that will describe variation in sedation practice prior to the trial across UK ICUs, in a separate publication. We believe this will provide important context for subsequent interpretation of the trial results.

If the patient is deeply sedated this would influence the time to successful extubation and the presence of coma and delirium free days. If the patient is deeply sedated the reason should be registered. As it is now it is up to the single doctor to decide if deep sedation is necessary. There should be some common rules to decide if deep sedation is necessary.

We agree that 'unnecessary' deep sedation could compromise the fidelity of the interventions, as per the comment above. However, in a pragmatic trial design this is difficult to control tightly. Further, it is not feasible to collect daily data for >1000 participants on reasons for deep sedation during every intervention day. However, we do ask the bedside caring nurse to record if there was a clinical indication for deep sedation. This is included in the 'nursing shift' data collection form that is completed by the caring nurse for each 12-hours nursing shift. Possible reasons for deep sedation are included in guidance, including brain injury, seizures, and requirement for advanced mechanical ventilation modes. These data are all captured in the trial database. We note this in on page 14 in the 'Management during the intervention period' section as follows: 'For each 12 hours nursing shift, clinical staff document whether there is a clinical indication for deep sedation. If deep sedation is required, the allocated sedative agent is titrated to achieve this if feasible.' We have added some additional clarification to this section in the revised manuscript.

The time to successful extubation is also influenced by the time sedation is stopped. In the protocol it should be described when to stop sedation. This would introduce a more uniform treatment between different sites.

Extubation is up to the discretion of the attending doctor. A more homogeneous treatment would be introduced if there were certain clinical endpoint which should be fulfilled before extubation.

These comments are linked so we address them together. As per the comments above, this is a large multicentre pragmatic effectiveness trial so it is neither feasible or our intention to closely control clinical decisions about when to wean and stop sedation and/or extubate the patient. To illustrate, we estimate that more than 800 different critical care consultants will care at some time for a patient in the trial. We expect differences in practice between clinicians will be balanced between groups by randomisation. If the allocated group influences when clinicians consider sedation cessation and/or extubation can be undertaken this is in part what we are testing in the trial, so we do not consider it an issue in relation to answering our trial questions. To make this clearer we have added the following to the section on 'weaning from mechanical ventilation' on page 15 as follows: 'The protocol does not control decisions about weaning sedation and mechanical ventilation tightly, given the pragmatic effectiveness design. Decisions and their timing are at the discretion of the responsible clinical team.'

What will happen if it is not possible to obtain sufficient sedation with clonidine or dexmedetomidine

and propofol is used to supplement the sedation. Will the patient be excluded?.

If clonidine or dexmedetomidine sedation is not sufficient it might be tempting to increase the administration of opioids. In this way the administration of opioids will be increased in these treatment groups. It should be stated that this must be avoided and the administration of opioids ought to be registered.

We agree that clarity on the way sedation is delivered, and the use of alternative and/or 'rescue' medications, is important. We included a specific section describing how this is done under the 'intervention groups' section on page 13. We believe the text we included clearly describes this, specifically the following section: 'Bedside clinical staff transition patients to achieve sedation with the allocated alpha2-agonist agent as quickly as clinically feasible and safe, using bedside guidance algorithms (see supplementary material). Additional opioid is used for analgesia using clinical judgement. Once alpha2-agonist is established, additional propofol is only recommended when the maximum alpha2-agonist dose is reached or because cardiovascular or other side-effects limit dose escalation.'

This section also references the detailed algorithms used by bedside nurses, which are included in the supplementary material. We note the point about doses of drugs, including opioids. We are capturing daily doses of all sedation drugs (including opioids), and any 'rescue medication' for agitation and delirium. These are entered into the trial database and will be available for reporting. Table 3 notes that daily drug use is being captured. In relation to withdrawal of patients, any requirement of additional propofol or other agents is not an indication for withdrawal. All patients are included post-randomisation on 'intention-to-treat'. The only withdrawal criteria are detailed in section on 'Withdrawals' on page 15. Although not directly relevant to the question from Dr Toft, we have a detailed description of various sensitivity analysis, and the Estimand for the trial, in the statistical analysis plan, which is included in the supplementary material.